# Anemia and Iron Deficiency Predict All-Cause Mortality in Patients with Heart Failure and Preserved Ejection Fraction: 6-Year Follow-Up Study

**DOI:** 10.3390/diagnostics14020209

**Published:** 2024-01-18

**Authors:** Fatoş Dilan Köseoğlu, Bülent Özlek

**Affiliations:** 1Department of Internal Medicine, Division of Hematology, Faculty of Medicine, Bakircay University, 35665 Izmir, Turkey; fatosdilankoseoglu@bakircay.edu.tr; 2Department of Cardiology, Faculty of Medicine, Mugla Sitki Kocman University, 48000 Mugla, Turkey

**Keywords:** anemia, iron deficiency, heart failure with preserved ejection fraction, mortality, prognosis

## Abstract

Aims: The aim of this study was to assess the prevalence of anemia and iron deficiency in patients with heart failure with preserved ejection fraction (HFpEF) and its impact on clinical outcomes. Methods: We retrospectively analyzed 212 patients with HFpEF and identified anemia as a serum hemoglobin level of less than 13 g/dL in men and less than 12 g/dL in women. Additionally, ID was defined as a serum ferritin concentration < 100 ng/mL or 100–299 ng/mL with transferrin saturation < 20%. Patients were followed up for an average of 66.2 ± 12.1 months, with the endpoint being all-cause mortality among patients with HFpEF, both with and without anemia and iron deficiency. Furthermore, we explored other predictors of all-cause mortality. Results: The average age of the entire group was 70.6 ± 10.5 years, with females comprising 55% of the patients. Anemia was present in 81 (38.2%) patients, while 108 (50.9%) had iron deficiency. At the end of the follow-up period, 60 (28.3%) of the patients had passed away. Patients with anemia displayed more heart failure (HF) symptoms, diastolic dysfunction, higher NT-pro-BNP levels, and worse baseline functional capacity than those without. Similarly, patients with iron deficiency showed more pronounced HF symptoms and worse functional capacity than those without. The results from the multivariable analyses revealed that anemia (hazard ratio [HR]: 5.401, 95% confidence interval [CI]: 4.303–6.209, log-rank *p* = 0.001), advanced age, iron deficiency (HR: 3.502, 95% CI: 2.204–6.701, log-rank *p* = 0.015), decreased left ventricular ejection fraction, chronic kidney disease, and paroxysmal nocturnal dyspnea were all independently associated with all-cause mortality. Conclusions: It is essential to consider anemia and iron deficiency as common comorbidities in managing and prognosis HFpEF, as they significantly increase mortality risk.

## 1. Introduction

Heart failure (HF) is a widespread and life-threatening public health issue that significantly burdens healthcare systems worldwide. The existing guidelines classify HF into three groups: HF with reduced ejection fraction (HFrEF), HF with mildly reduced ejection fraction (HFmrEF), and HF with preserved ejection fraction (HFpEF) [1]. Epidemiological studies indicate that more than 64 million people worldwide have HF, with nearly half being patients with HFpEF [2]. The importance of HFpEF has become more apparent in recent years as diagnostic modalities for HFpEF have become increasingly clear, and awareness of this type of heart failure has increased among clinicians. Although risk factors, prognostic markers, and treatment modalities have been defined for HFrEF, there is still uncertainty regarding these issues in HFpEF.

Comorbidities can significantly impact the survival and prognosis of individuals with HF. Many comorbidities have been shown to increase mortality and morbidity in patients with HF [3]. Anemia and iron deficiency (ID) are common pathologies that require management in clinical practice, similar to HFpEF. One of the leading causes of anemia globally is ID [4]. Anemia or ID’s presence can lead to poor clinical outcomes in various cardiovascular and non-cardiovascular conditions such as HFrEF, ischemic heart disease, chronic kidney disease, inflammatory bowel diseases, and cancer [5,6,7,8,9]. HF guidelines recommend iron therapies and target values in patients with HFmrEF and HFrEF [1,5]. However, there are insufficient data to form a clear opinion on the prognostic importance of anemia and ID in patients with HFpEF, who comprise a significant portion of the HF population. A better understanding of how these two common conditions, frequently found with HFpEF, affect clinical outcomes may lead to new treatment research. Therefore, our objective was to evaluate the effect of anemia and ID on long-term survival. We conducted long-term follow-ups on a cohort of patients with HFpEF.

## 2. Methods 

### 2.1. Study Design and Participants

This study was an observational and retrospective analysis conducted at a single center. The cohort consisted of 228 adult patients with HFpEF admitted to the cardiology outpatient clinic between September 2017 and April 2018. Anonymous clinical findings, comorbidities, treatments, demographics, laboratory tests, electrocardiogram, and echocardiographic data were thoroughly evaluated. However, the final analysis excluded 16 patients with incomplete baseline or follow-up data. 

According to the criteria of the World Health Organization, anemia is defined as a serum hemoglobin (Hb) level of less than 13 g/dL in men and less than 12 g/dL in women [10]. In patients with HFpEF, ID was defined as either a serum ferritin concentration < 100 ng/mL or 100–299 ng/mL with transferrin saturation (TSAT) < 20%. The diagnosis of HFpEF was made according to the “2021 European Society of Cardiology Guidelines for the Diagnosis and Treatment of Heart Failure” [1]. The following criteria were considered for the diagnosis of HFpEF: patients with New York Heart Association (NYHA) class I–IV symptoms, LVEF ≥ 50% on echocardiogram, evidence of structural or functional cardiac abnormalities such as left ventricular hypertrophy and/or left atrial enlargement and/or diastolic dysfunction, and N-terminal pro-brain natriuretic peptide (NT-pro-BNP) levels > 125 pg/mL with sinus rhythm and >365 pg/mL with atrial fibrillation (AF). Patients with primary severe heart valve disease requiring intervention or surgery; patients with any history of surgically corrected heart valve diseases (e.g., mechanical or bioprosthetic heart valves); patients with myocardial infarction, major bleeding, stroke, or coronary artery bypass graft surgery in the past 90 days; known infiltrative or hypertrophic obstructive cardiomyopathy or known pericardial constriction; patients with congenital heart diseases or cor pulmonale; and pregnant patients were excluded to be able to include “true” HFpEF patients in the study.

The estimated glomerular filtration rate (eGFR) was calculated using the Modification of Diet in Renal Disease equation. Chronic kidney disease (CKD) was defined as eGFR < 60 mL/min/1.73 m^2^. Hypertension was defined as systolic blood pressure > 140 mmHg or diastolic blood pressure > 90 mmHg, documented history of hypertension, or current use of antihypertensive medications. Coronary artery disease was defined as a physician-documented history of coronary artery disease, known coronary stenosis > 50%, history of myocardial infarction, percutaneous coronary intervention, coronary artery bypass grafting, or abnormal stress test results consistent with myocardial ischemia. Hyperlipidemia was defined as a physician-documented history of hyperlipidemia or current use of lipid-lowering medications. Peripheral artery disease, cerebrovascular accident, chronic obstructive pulmonary disease, depression, and malignancy were defined as the presence of documented history.

### 2.2. Follow-Up and Endpoints

The follow-up period for this study concluded on 10 September 2023, with the endpoint defined as all-cause mortality among patients with HFpEF, both with and without anemia and ID. Additionally, further analysis was conducted to assess the association between anemia, ID, and clinical outcomes. Patient status (dead or alive), date, and cause of death were obtained by querying the Ministry of Health’s Death Notification System and Mugla Training and Research Hospital’s medical database. 

This study followed the declaration of Helsinki, and all study protocols were approved by the institutional review board of Mugla Sitki Kocman University (approval date and number: 02.10.2023/94). As the study was retrospective, informed consent was waived.

### 2.3. Statistical Analyses

To assess the normal distribution of variables, we utilized the Kolmogorov–Smirnov and Shapiro–Wilk tests. Continuous baseline variables were presented as mean ± standard deviations or median and interquartile range, based on data distribution. Categorical variables were expressed in frequencies and percentages. We compared continuous variables using either a Student *t*-test or Mann–Whitney U test and categorical variables using either a χ^2^ test or Fisher’s exact test. Our analysis included comparisons of baseline clinical and laboratory characteristics for deceased vs. surviving patients, anemic vs. non-anemic patients, and ID vs. non-ID patients. To explore the links between factors and overall mortality, we conducted a multivariable logistic regression analysis. In the study, factors that showed a significant association with mortality between the deceased and survivor groups with a *p*-value less than 0.05 in univariate analysis were selected for inclusion in the multivariable analysis. This approach was taken to identify potential risk factors that may have contributed to the difference in mortality rates between the two groups. The correlation between the parameters in the model was assessed by Pearson analysis. Potential risk factors that were correlated with each other were not included in the multivariable logistic regression analysis. Since Hb is correlated with anemia, ferritin with ID, ankle swelling and JVD with PND, these parameters were not included in the multivariable logistic regression analysis. The goal was to determine which factors remained significant after accounting for potential confounding variables in the multivariable analysis. For the outcome of all-cause mortality, we used multivariable logistic regression models while controlling for age, NYHA class, paroxysmal nocturnal dyspnea (PND), cardiac murmur, anemia, ID, CKD, malignancy, LVEF, E/e’ ratio, NT-pro-BNP, and blood urea nitrogen (BUN). We generated survival curves using the Kaplan–Meier approach and compared them between patients with anemia and ID vs. those without, utilizing the log-rank test. Statistical significance was defined as a 2-sided *p* < 0.05. All analyses were performed using SPSS 24.0 (IBM Corp, New York, NY, USA). 

## 3. Results

The research examined a cohort of 212 individuals diagnosed with HFpEF, whose average age was 70.6 ± 10.5 years, of which 55% were female. Of this group, 81 (38.2%) were found to have anemia and 108 (50.9%) had ID. Over the course of a follow-up period of approximately 66.2 ± 12.1 months, 60 patients passed away, resulting in a mortality rate of 28.3%. 

### 3.1. Baseline Clinical Characteristics of Deceased and Surviving Patients

Clinical information, laboratory results, and treatment details for surviving and deceased patients are provided in Table 1 and Table 2. The deceased patients were, on average, 76 years old, which is higher than the average age of survivors at 68 years (*p* < 0.001). They also exhibited more symptoms and signs such as orthopnea (21.7 vs. 7.9%, *p* = 0.005), PND (63.3 vs. 39.5%, *p* = 0.002), ankle swelling (33.3 vs. 14.5%, *p* = 0.002), JVD (31.7 vs. 15.1%, *p* = 0.007), cardiac murmur (78.3 vs. 59.2%, *p* = 0.009), peripheral edema (33.3 vs. 13.2%, *p* = 0.001), pulmonary crepitations (11.7 vs. 3.9%, *p* = 0.035), and tachypnea (11.7 vs. 2.6%, *p* = 0.008). As expected, dying patients had worse NYHA capacity (*p* = 0.002). Interestingly, there were no significant differences between the two groups in terms of smoking and alcohol use, blood pressure, body mass index, and heart rate. Moreover, there was no significant difference in the prevalence of various diseases, including AF, hypertension, diabetes mellitus, obstructive sleep apnea, hyperlipidemia, coronary artery disease, peripheral artery disease, cerebrovascular accident, chronic obstructive pulmonary disease, or depression. However, the prevalence of anemia (73.3 vs. 24.3%, *p* < 0.001), ID (80 vs. 39.5%, *p* < 0.001), CKD (25 vs. 6.6%, *p* < 0.001), and malignancy (6.7 vs. 1.3%, *p* = 0.034) was higher in the deceased patients. Additionally, the follow-up period was shorter for the deceased patients, averaging 51.5 months compared to 68.6 months for survivors (*p* < 0.001).

We examined the detailed echocardiographic findings of the two groups and found that patients who died had more pronounced diastolic dysfunction (E/e’ = 12.3 in deceased and 11 in alive, *p* = 0.006). Although the LVEF was above 50% in all patients, it was statistically lower in patients who died (55.8 vs. 57.6%, *p* = 0.005) than in survivors. However, there was no significant difference between non-survivors and survivors regarding left atrial enlargement, left ventricular hypertrophy, systolic pulmonary artery pressure, or non-critical valvular diseases.

Blood test results also showed significant differences between patients who survived and those who did not. Those who passed away had higher levels of NT-pro-BNP (895 vs. 454 pg/mL, *p* = 0.017), BUN (21.2 vs. 17.9 mg/dL, *p* = 0.010), and serum creatinine (1.0 vs. 0.9 mg/dL, *p* = 0.049) than those who survived. Conversely, non-survivors had lower levels of Hb (11.1 vs. 13.5 g/dL, *p* < 0.001), TSAT (12.5 vs. 35.2%, *p* < 0.001), serum ferritin (34.1 vs. 132.5 ng/mL, *p* < 0.001), and calcium (9.0 vs. 9.3 mg/dL, *p* = 0.009). However, no significant differences between the two groups were found when drug treatments were analyzed. 

### 3.2. Baseline Clinical Characteristics by the Presence of Anemia

The clinical characteristics of patients were compared in Table 3 based on their anemia status. The study found that patients with anemia tended to be older, with an average age of 74 years, compared to those without anemia, who had an average age of 68 years (*p* < 0.001). Patients with anemia also displayed symptoms such as PND (64.2 vs. 35.1%, *p* < 0.001), bendopnea (39.5 vs. 19.8%, *p* = 0.002), and pulmonary crepitations (12.3 vs. 2.3%, *p* = 0.003) more frequently with poorer NYHA functional capacity (*p* < 0.001). Additionally, patients with anemia had a higher prevalence of ID (90.1 vs. 26.7%, *p* < 0.001), CKD (19.8 vs. 6.9%, *p* = 0.005), and malignancy (6.2 vs. 0.8%, *p* = 0.021). However, there was no significant difference in other comorbid diseases. Echocardiography also revealed that anemic patients had more diastolic dysfunction (E/e’ 12.1 vs. 10.9 in non-anemics, *p* = 0.008) and left atrial enlargement (67.9 vs. 54.2%, *p* = 0.048). Blood tests showed that Hb (11 vs. 13.9 g/dL, *p* < 0.001), TSAT (15.5 vs. 37%, *p* < 0.001), and serum ferritin (35 vs. 141 ng/mL, *p* < 0.001) levels were lower in the frail group. Serum calcium was also insufficient (9.0 vs. 9.3 mg/dL, *p* < 0.001) in patients with anemia compared to non-anemic patients. On the other hand, patients with anemia had higher levels of NT-pro-BNP (1041 vs. 430 pg/mL, *p* = 0.002), BUN (21.1 vs. 17.3 mg/dL, *p* = 0.001), and serum creatinine (1.0 vs. 0.9 mg/dL, *p* = 0.036). The only difference in medical treatments between the two groups was the rate of oral antidiabetic use. 

It is worth noting that all-cause mortality rates were significantly higher among patients with anemia than those without anemia (54.3 vs. 12.2%, *p* < 0.001). Patients without anemia had a longer follow-up period due to their lower mortality rate (68.7 vs. 57.8 months, *p* < 0.001). 

### 3.3. Baseline Clinical Characteristics by the Presence of ID

Table 4 outlines the clinical and laboratory features of patients, categorized by those with and without ID. Patients with ID had a higher prevalence of PND (56.5 vs. 35.6%, *p* = 0.002) and were generally older (73.6 vs. 67.6 years, *p* < 0.001) with poorer NYHA functional capacity (*p* = 0.010). Although the two groups did not differ significantly in comorbid diseases, patients with ID had a higher frequency of anemia (67.6 vs. 7.7%, *p* < 0.001). The echocardiographic findings were comparable between the two groups. Blood test analysis revealed that patients with ID had lower Hb levels (11.9 vs. 13.8 g/dL, *p* < 0.001), TSAT (18 vs. 40%, *p* < 0.001), and serum ferritin (36.5 vs. 209.5 ng/mL, *p* < 0.001). The medical treatment used for patients with and without ID was similar.

Patients with ID were observed for an average of 61.2 months, while those without ID were followed up for 69.4 months. Notably, patients with ID had significantly higher all-cause mortality rates (44.4 vs. 11.5%, *p* < 0.001). 

### 3.4. Cardiovascular and Non-Cardiovascular Mortality Rates by the Presence of Anemia or ID 

Within the group, 26 patients died from cardiovascular causes, while 34 patients died from non-cardiovascular causes such as sepsis, cancer, bacterial pneumonia, and COVID-19. Interestingly, the mortality rate for cardiovascular reasons was quite similar in both groups, with 47.7% (21 out of 44) in the anemic HFpEF group and 45.8% (22 out of 48) in the HFpEF patients with ID group, as illustrated in Figure 1. Patients with anemia or ID had higher rates of both cardiovascular and non-cardiovascular mortality than those without (*p* < 0.001).

### 3.5. Independent Predictors of All-Cause Mortality

The data in Table 5 display the factors that predict all-cause mortality for the HFpEF group during long-term follow-up. Upon univariable analysis of the data, mortality was found to be linked to increasing age, poor NYHA functional class, PND, cardiac murmur, anemia, ID, CKD, malignancy, decreased LVEF, increased NT-pro-BNP, E/e’, BUN. A multivariable logistic regression model was then created to eliminate potential confounders. After adjustment, the multivariable analyses showed that anemia (hazard ratio [HR]: 5.401, 95% confidence interval [CI]: 4.303–6.209, *p* = 0.001), advanced age (HR: 1.103, 95% CI: 1.063–1.182, *p* = 0.002), ID (HR: 3.502, 95% CI: 2.204–6.701, *p* = 0.015), decreased LVEF (HR for increasing: 0.837, 95% CI: 0.725–0.974, *p* = 0.016), CKD (HR: 2.166, 95% CI: 1.500–3.825, *p* = 0.020), and PND (HR: 1.421, 95% CI: 1.097–2.508, *p* = 0.27) were all independently associated with all-cause death. Additionally, Figure 2 illustrates the Kaplan–Meier survival curves of HFpEF patients with and without anemia, while Figure 3 illustrates the Kaplan–Meier survival curves of HFpEF patients with and without ID.

## 4. Discussion

The present study offers valuable insights into predicting the prognosis of patients with HFpEF. Notable findings include: (i) The HFpEF group had a mortality rate of 28.3% over an average follow-up of 5.5 years; (ii) 38.2% of patients had anemia and 50.9% had ID; (iii) advanced age, anemia, ID, and CKD were all independent indicators of all-cause mortality in HFpEF; (iv) Patients with both anemia and HfpEF had a five-fold greater relative risk of death compared to those without anemia; (v) HfpEF patients with ID had significantly higher mortality rates than those without ID.

Numerous studies have been conducted to determine the demographic and clinical traits of those suffering from HfpEF [11,12]. However, the intricacies of this condition pose several hurdles, such as comprehending its pathophysiology, developing successful methods to prolong life, and creating ways to diagnose it. These factors make HFpEF a multifaceted and complex ailment that necessitates a thorough comprehension of its underlying mechanisms to formulate effective treatment plans. Recently, research has indicated that the use of sodium-glucose co-transporter 2 inhibitors can decrease mortality rates [13], while glucagon-like peptide-1 analogs have been shown to improve symptoms [14]. These findings have led to the possibility that HFpEF could be considered a metabolic disease [15]. It appears that the prevailing drug therapies that have proven successful in addressing HFrEF may not yield comparable outcomes when applied to HFpEF treatment [16,17,18]. This implies that there may be other factors at play in the development of HFpEF beyond just pump failure. Therefore, it is becoming more critical to explore how various comorbid conditions could impact the outlook for those with HFpEF.

Anemia is a commonly occurring comorbid condition in patients with HFpEF. Although anemia in premenopausal women may be due to physiological blood loss, it may also be a sign of comorbid diseases or treatments that reduce erythropoiesis or trigger blood loss, especially in the elderly [19]. According to previous large observational studies, anemia is prevalent in 22 to 41% of the HFpEF population [12,20]. Our study found that the rate of anemia in the HFpEF cohort was 38.2%, which is consistent with these data. It is essential to address the association of HFpEF and anemia regarding disease prognosis and physiopathology. A recently published analysis of approximately 90,000 patients showed that the presence of anemia is an independent risk factor for the development of HFpEF [21]. While it is not yet confirmed, it appears that the physiology of HFpEF increases the risk of anemia, and the presence of anemia facilitates the occurrence of HFpEF. Only a few studies have examined the impact of anemia on HFpEF’s clinical outcomes. A post hoc analysis of the “Treatment of Preserved Cardiac Function Heart Failure with an Aldosterone Antagonist” study followed up with 1748 HFpEF patients for a median of 2.4 years [22], revealing a mortality rate of 22% in the entire cohort, similar to the all-cause mortality rates in our study. Anemia emerged as an independent predictor of all-cause mortality and HF hospitalizations at the end of the follow-up. Furthermore, an interesting finding emerged from the study. According to the results, sudden cardiac deaths, thought to be of cardiovascular origin, were higher in anemic HFpEF patients. At the same time, no difference was found between anemic and non-anemic patients in fatalities due to pump failure [22]. This discovery reinforces the idea that poor outcomes in HFpEF cannot be solely attributed to pump failure. In our study, patients with anemia had higher cardiovascular and non-cardiovascular mortality rates than those without anemia. According to the Swedish-HF study [23], which examined around 10,000 patients with HFpEF, the 1-year all-cause mortality rate was 43% in anemic patients. In our analysis, this rate was 54.3%. Given that our follow-up period spanned over 5 years, it is reasonable to view the results as comparable. Once again, anemia was identified as a risk factor for all-cause mortality in the Swedish-HF study [23], lending support to our findings. In a comprehensive meta-analysis of 28,735 patients by Majmundar et al., anemia was found to be a prognostic factor for all-cause mortality and all-cause hospitalizations in patients with HFpEF [24]. Furthermore, other studies have shown that anemia substantially raises mortality rates in patients with HFpEF [25,26,27]. The importance of long-term survival and its related factors in patient care cannot be overstated. However, it is worth noting that reduced exercise capacity can also significantly impact the quality-of-life for these individuals. Although our study did not directly measure this, the prevalence of heart failure symptoms and diastolic dysfunction in anemic patients suggests that this is a valid concern. A recent exercise stress echocardiography-based study by Naito et al. found that anemic HFpEF patients experience more severe effects on exercise capacity and oxygen consumption compared to non-anemic patients, with respiratory inefficiency being particularly concerning during peak exercise [28]. It has been established that anemic patients experiencing chemoreceptor stimulation due to hypoxia tend to have higher heart rates as a result of sympathetic system activation [29]. However, our study revealed no statistically significant difference in the average heart rate between anemic and non-anemic patients in our patient population. This outcome may be attributed to several factors, such as the relatively small sample size, the use of rate-decelerating treatment agents, and the absence of severe anemia (Hb < 7 g/dL) among the patients.

It has been estimated that between 47–68% of chronic HF patients suffer from ID, with a slightly higher prevalence in HFpEF [5]. Our study found a 50.9% prevalence rate of ID in the HFpEF cohort. Factors such as decreased iron absorption due to HF-related diffuse edema, nutritional deficiencies, reduced release of stored iron, and blood loss through the gastrointestinal tract may contribute to ID in these patients, along with inflammation’s role in HF [30]. ID can affect oxygen consumption pathways through various mechanisms, including reducing oxygen storage and carrying capacity and causing mitochondrial dysfunction in both heart and skeletal muscle. This leads to reduced energetic efficiency and anaerobic metabolism [31]. It is crucial to consider the significance of ID, which has been shown to have prognostic and quality-of-life implications in HFrEF [32]. In a study of 190 HFpEF patients, Bekfani et al. found that ID was linked to worse outcomes in the 6-min walk test, cardiopulmonary exercise test, and quality-of-life assessment scale [33]. Similarly, another study discovered that ID was associated with poor functional capacity and decreased quality-of-life in HFpEF [34]. A meta-analysis of 1424 HFpEF patients showed that the prevalence of ID was 59%, with ID being linked to worse exercise capacity and functional outcomes but not hospitalization or mortality [35]. However, there is a lack of large-scale research that can definitively demonstrate the prognostic significance of ID in this patient population. In Aizpurua et al.’s prospective study of 300 HFpEF patients, ID was significantly correlated with a 3.5-fold increase in all-cause mortality but not HF hospitalizations [36]. Our study found that HFpEF patients with ID had a 3.5-fold higher relative risk of all-cause mortality compared to those without ID at a follow-up of 5.5 years. However, in a separate study of 448 HFpEF patients with a 2-year follow-up, ID was not found to be predictive of all-cause mortality or the combined endpoint of death/HF hospitalization [37]. The shorter follow-up period in this study may explain why ID was not identified as a risk factor for mortality. Given the heterogeneous nature of HFpEF, it may be more appropriate to obtain more accurate data over longer follow-up periods.

### Study Limitations and Strengths

Our study has some anticipated limitations. Of utmost significance is the fact that it was conducted retrospectively at a single center. Additionally, the number of participants was relatively small. However, given the scarcity of studies with larger patient pools that specifically examine the correlation between ID and HFpEF, we are confident that the number of patients we included is substantial enough to yield reliable and practical data. Another limitation we encountered was our inability to analyze whether patients received iron supplementation or other anemia treatments during the follow-up period.

This study boasts several notable strengths, including one of the longest follow-up periods in the literature on HFpEF, anemia and ID, as well as the comprehensive nature of its echocardiographic and laboratory data. By incorporating numerous data points into the multivariable regression analysis model, potential confounders are effectively eliminated, resulting in reliable and significant values.

## 5. Conclusions

According to this study, a minimum of 50% of patients with HFpEF experience ID, while over a third suffer from anemia. In addition, the observed all-cause mortality rate for this group of patients was 28.3% over a follow-up duration of about 5.5 years. Anemic patients were found to have a relative risk of death approximately 5-times higher, while patients with ID had a relative risk of about 3.5-times higher. Furthermore, advanced age and CKD are additional indicators of all-cause mortality in HFpEF patients. Given the complex characteristics of the disease and the uncertainties surrounding the survival, prognostic predictors, and new treatment modalities of HFpEF, it is crucial to investigate the impact of anemia and ID, which are prevalent among patients, on this syndrome. This issue warrants further examination both in theory and clinical practice. As such, it is recommended that comprehensive, prospective, and multicenter studies be conducted to corroborate the results of our research.

## Figures and Tables

**Figure 1 diagnostics-14-00209-f001:**
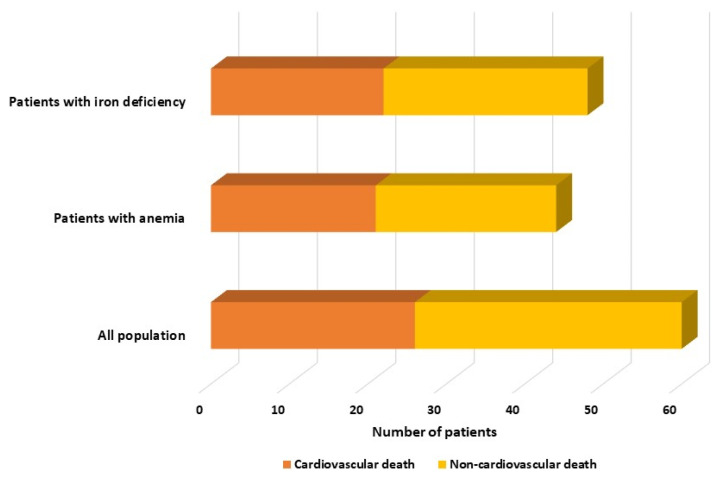
Distribution of mortality rates among the whole cohort, anemic, and iron-deficient patients, categorized by cardiovascular and non-cardiovascular causes.

**Figure 2 diagnostics-14-00209-f002:**
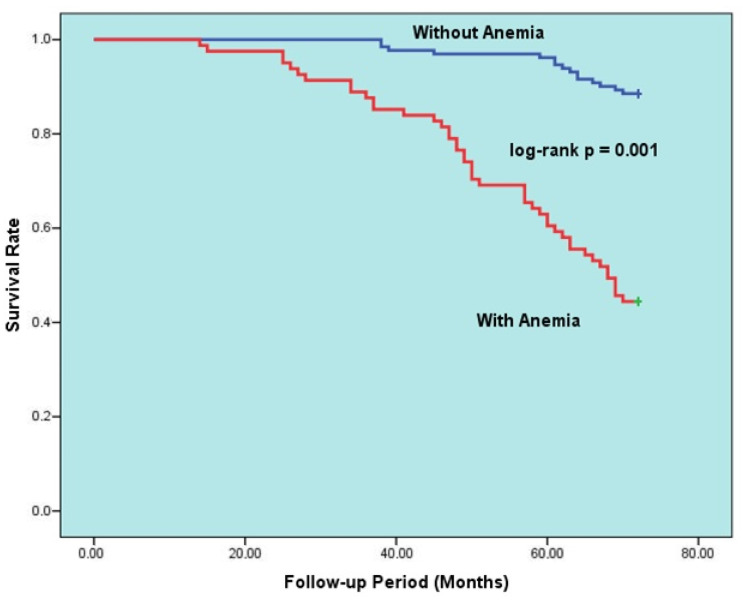
Kaplan–Meier 6-year survival rates of HFpEF patients with and without anemia.

**Figure 3 diagnostics-14-00209-f003:**
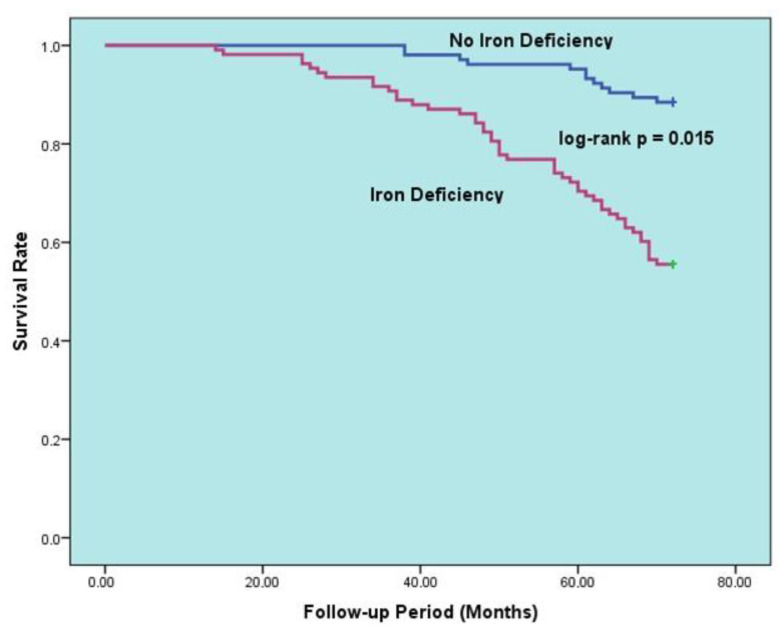
Kaplan–Meier curves for survival in patients with HFpEF according to presence vs. absence of iron deficiency.

**Table 1 diagnostics-14-00209-t001:** Clinical characteristics of survivor and non-survivor patients.

		Overall (*n* = 212)	Deceased (*n* = 60)	Surviving (*n* = 152)	*p*-Value
Female sex, *n* (%)		116 (54.7)	37 (61.7)	79 (52)	0.202
Age, years		70.6 ± 10.5	76.3 ± 8.4	68.4 ± 10.5	<0.001
Smoking, *n* (%)		35 (16.5)	7 (11.7)	28 (18.4)	0.233
Alcohol use, *n* (%)		9 (4.2)	0 (0)	9 (5.9)	0.054
Place of residence	Rural	109 (51.4)	34 (56.7)	75 (49.3)	0.157
	Urban	103 (48.6)	25 (41.7)	77 (50.7)	
NYHA	I–II	175 (82.5)	42 (70)	133 (87.5)	0.002
	III–IV	37 (17.5)	18 (30)	19 (12.5)	
Orthopnea, *n* (%)		25 (11.8)	13 (21.7)	12 (7.9)	0.005
Paroxysmal nocturnal dyspnea, *n* (%)		98 (46.2)	38 (63.3)	60 (39.5)	0.002
Bendopnea, *n* (%)		58 (27.4)	21 (35)	37 (24.3)	0.117
Palpitation, *n* (%)		95 (44.8)	27 (45)	68 (44.7)	0.972
Reduced exercise tolerance, *n* (%)		201 (94.8)	57 (95)	144 (94.7)	0.938
Fatigue, tiredness, *n* (%)		167 (78.8)	52 (86.7)	115 (75.7)	0.077
Ankle swelling, *n* (%)		42 (19.8)	20 (33.3)	22 (14.5)	0.002
Chest pain, *n* (%)		28 (13.2)	8 (13.3)	20 (13.2)	0.973
Body mass index, kg/m^2^		28.2 ± 4.6	27.7 ± 4.1	28.4 ± 4.7	0.378
Systolic blood pressure, mmHg		129.6 ± 15	130.1 ± 16	129.4 ± 15	0.224
Diastolic blood pressure, mmHg		78.7 ± 10	79.3 ± 9.8	78.5 ± 10.2	0.301
Heart rate, bpm		82 ± 16	79 ± 13	83 ± 17	0.082
Jugular vein distention, *n* (%)		42 (19.8)	19 (31.7)	23 (15.1)	0.007
Cardiac murmur, *n* (%)		137 (64.6)	47 (78.3)	90 (59.2)	0.009
Third heart sound, *n* (%)		9 (4.2)	4 (6.7)	5 (3.3)	0.272
Peripheral edema, *n* (%)		40 (18.9)	20 (33.3)	20 (13.2)	0.001
Pulmonary crepitations, *n* (%)		13 (6.1)	7 (11.7)	6 (3.9)	0.035
Tachypnea, *n* (%)		11 (5.2)	7 (11.7)	4 (2.6)	0.008
ECG abnormality, *n* (%)		110 (51.9)	32 (53.3)	78 (51.3)	0.791
Ascites, *n* (%)		1 (0.5)	1 (1.7)	0 (0)	0.111
Cachexia, *n* (%)		4 (1.9)	1 (1.7)	3 (2)	0.882
Comorbidities, *n* (%)					
Anemia		81 (38.2)	44 (73.3)	37 (24.3)	<0.001
Iron deficiency		108 (50.9)	48 (80)	60 (39.5)	<0.001
Atrial fibrillation		83 (39.2)	25 (41.7)	58 (38.2)	0.637
Hypertension		163 (76.9)	46 (76.7)	117 (77)	0.962
Diabetes mellitus		66 (31.1)	21 (35)	45 (29.6)	0.445
Chronic kidney disease		25 (11.8)	15 (25)	10 (6.6)	<0.001
Dialysis		2 (0.9)	2 (3.3)	0 (0)	0.024
Obstructive sleep apnea		25 (11.8)	3 (5)	22 (14.5)	0.054
Hyperlipidemia		81 (38.2)	22 (36.7)	59 (38.8)	0.772
Coronary artery disease		70 (33)	18 (30)	52 (34.2)	0.557
Coronary artery by-pass grafting		21 (9.9)	7 (11.7)	14 (9.2)	0.590
Percutaneous coronary intervention		36 (17)	7 (11.7)	29 (19.1)	0.195
Peripheral artery disease		8 (3.8)	2 (3.3)	6 (3.9)	0.833
CVA/TIA		15 (7.1)	7 (11.7)	8 (5.3)	0.101
COPD		31 (14.6)	10 (16.7)	21 (13.8)	0.597
Hepatic failure		0 (0)	0 (0)	0 (0)	-
Depression		18 (8.5)	5 (8.3)	13 (8.6)	0.959
Malignancy		6 (2.8)	4 (6.7)	2 (1.3)	0.034
Follow-up period, months		66.2 ± 12.1	51.5 ± 14.9	68.6 ± 10.4	<0.001

Data are presented as mean ± standard deviation or median and interquartile range or number (%). Abbreviations: COPD, chronic obstructive pulmonary disease; CVA, cerebrovascular accident; ECG, electrocardiogram; NYHA, New York Heart Association; TIA, transient ischemic attack.

**Table 2 diagnostics-14-00209-t002:** Echocardiographic data, blood tests, and medications of deceased and surviving patients.

		Overall (*n* = 212)	Deceased (*n* = 60)	Surviving (*n* = 152)	*p*-Value
Echocardiographic data					
LVEF, %		57 ± 4.2	55.8 ± 3.8	57.6 ± 4.2	0.005
e’, cm/sn		6.8 ± 1.9	6.6 ± 1.9	6.9 ± 1.9	0.203
E/e’		11.3 ± 3.2	12.3 ± 3.8	11 ± 2.9	0.006
LAVI, mL/m^2^		37.5 ± 9.6	38.6 ± 10	37.1 ± 9.5	0.298
LA enlargement, *n*(%)		126 (59.4)	40 (66.7)	86 (56.6)	0.178
LVMI, g/m^2^		111.6 ± 24.1	114.1 ± 25.3	110.7 ± 23.7	0.369
LV concentric hypertrophy, *n* (%)		131 (61.8)	40 (66.7)	91 (59.9)	0.359
PAPs, mmHg		22.7 ± 10.7	24.4 ± 11.6	22 ± 10.3	0.173
Mitral regurgitation	None	49 (23.1)	10 (16.7)	39 (25.7)	0.259
	Mild	120 (56.6)	39 (65)	81 (53.3)	
	Moderate	43 (20.3)	11 (18.3)	32 (21.1)	
Mitral stenosis	None	194 (91.5)	58 (96.7)	146 (96.1)	0.833
	Mild	8 (8.5)	2 (3.3)	6 (3.9)	
	Moderate	0 (0)	0 (0)	0 (0)	
Aortic stenosis	None	197 (92.9)	56 (93.3)	141 (92.8)	0.820
	Mild	14 (6.6)	4 (6.7)	10 (6.6)	
	Moderate	1 (0.5)	0 (0)	1 (0.7)	
Aortic regurgitation	None	161 (75.9)	42 (70)	119 (78.3)	0.132
	Mild	43 (20.3)	17 (28.3)	26 (17.1)	
	Moderate	8 (3.8)	1 (1.7)	7 (4.6)	
Tricuspid regurgitation	None	110 (51.9)	30 (50)	80 (52.6)	0.509
	Mild	66 (31.1)	17 (28.3)	49 (32.2)	
	Moderate	36 (17)	13 (21.7)	23 (15.1)	
Blood tests					
Hemoglobin, g/dL		12.8 ± 1.9	11.1 ± 1.5	13.5 ± 1.6	<0.001
Transferrin saturation, mean (%)		28.8	12.5	35.2	<0.001
Ferritin, ng/mL		94.5 (35–207)	34.1 (20–64.5)	132.5 (67–244.5)	<0.001
NT-proBNP, pg/mL		568.5 (278–1339)	895 (431.5–1436.5)	454 (266–1100.5)	0.017
Fasting blood glucose, mg/dL		111.4 ± 37	107.7 ± 35.7	112.9 ± 37.6	0.361
BUN, mg/dL		18.8 ± 8.1	21.2 ± 8.6	17.9 ± 7.8	0.010
Serum Creatinine, mg/dL		0.9 ± 0.3	1 ± 0.5	0.9 ± 0.4	0.049
Serum Sodium, mmol/L		140.6 ± 10	139.8 ± 10.2	141 ± 9.9	0.548
Serum Potassium, mmol/L		4.5 ± 0.5	4.5 ± 0.5	4.5 ± 0.5	0.538
Serum Calcium, mg/dL		9.2 ± 0.5	9 ± 0.5	9.3 ± 0.5	0.009
Uric acid, mg/dL		5.9 ± 1.4	6.2 ± 1.7	5.8 ± 1.4	0.060
Leukocyte, ×10^3^/µL		8 ± 2.3	8 ± 2.3	8.1 ± 2.4	0.706
C-reactive protein, mg/dL		4 (1.7–8)	3.4 (1.4–9.3)	4 (2–7.1)	0.315
TSH, µIU/mL		1.2 (0.8–1.9)	1.2 (0.7–1.9)	1.2 (0.9–1.9)	0.307
Medications, n(%)					
Angiotensin-converting enzyme inhibitor		56 (26.4)	19 (31.7)	37 (24.3)	0.276
Angiotensin receptor blocker		55 (25.9)	12 (20)	43 (28.3)	0.215
Beta-blocker		110 (51.9)	30 (50)	80 (52.6)	0.730
Aldosterone antagonists		29 (13.6)	7 (11.7)	22 (14.5)	0.592
Amiodarone		4 (1.8)	2 (3.3)	2 (1.3)	0.331
Nondihydropyridine calcium blockers		24 (11.3)	6 (10)	18 (11.8)	0.703
Dihydropyridine calcium blockers		45 (21.2)	16 (26.7)	29 (19.1)	0.224
Digoxin		18 (8.5)	5 (8.3)	13 (8.6)	0.959
Statin		55 (25.9)	14 (23.3)	41 (27)	0.586
Loop diuretic		45 (21.2)	13 (21.7)	32 (21.1)	0.922
Thiazide		67 (31.6)	20 (33.3)	47 (30.9)	0.734
Isosorbide		6 (2.8)	1 (1.7)	5 (3.3)	0.521
Antiaggregant		68 (32.1)	21 (35)	47 (30.9)	0.567
Anticoagulant		71 (33.5)	25 (41.7)	46 (30.3)	0.113
Nonsteroidal anti-inflammatory drug		29 (13.7)	8 (13.3)	21 (13.8)	0.927
Oral antihyperglysemic		57 (26.9)	21 (35)	36 (23.7)	0.094
Insulin		13 (6.1)	3 (5)	10 (6.6)	0.666

Data are presented as mean ± standard deviation or median and interquartile range or number (%). Abbreviations: BUN, blood urea nitrogen; LA, left atrium; LAVI, left atrial volume index; LV, left ventricle; LVEF, left ventricle ejection fraction; LVMI, left ventricular mass index; NT-proBNP, N-terminal pro-B-type natriuretic peptide; PAPs, pulmonary artery systolic pressure; TSH, thyrotropin stimulating hormone.

**Table 3 diagnostics-14-00209-t003:** Clinical characteristics of patients with and without anemia.

		Overall (*n* = 212)	With Anemia (*n* = 81)	Without Anemia (*n* = 131)	*p*-Value
Female sex, *n* (%)		116 (54.7)	47 (58)	69 (52.7)	0.447
Age, years		70.6 ± 10.5	74.4 ± 8.7	68.3 ± 11	<0.001
NYHA	I–II	175 (82.5)	57 (70.4)	118 (90.1)	<0.001
	III–IV	37 (17.5)	24 (29.6)	13 (9.9)	
Paroxysmal nocturnal dyspnea, *n* (%)		98 (46.2)	52 (64.2)	46 (35.1)	<0.001
Bendopnea, *n* (%)		58 (27.4)	32 (39.5)	26 (19.8)	0.002
Reduced exercise tolerance, *n* (%)		201 (94.8)	78 (96.3)	123 (93.9)	0.443
Body mass index, kg/m^2^		28.2 ± 4.6	28.3 ± 4.6	28.1 ± 4.6	0.751
Systolic blood pressure, mmHg		129.6 ± 15	130.6 ± 15.3	129 ± 15.2	0.445
Diastolic blood pressure, mmHg		78.7 ± 10	79.4 ± 10.5	78.3 ± 9.8	0.437
Heart rate, bpm		82 ± 16	80 ± 14	83 ± 17	0.185
Jugular vein distention, *n* (%)		42 (19.8)	21 (25.9)	21 (16)	0.079
Peripheral edema, *n* (%)		40 (18.9)	20 (24.7)	20 (15.3)	0.088
Pulmonary crepitations, *n* (%)		13 (6.1)	10 (12.3)	3 (2.3)	0.003
ECG abnormality, *n* (%)		110 (51.9)	41 (50.6)	69 (52.7)	0.771
Comorbidities, *n* (%)					
Iron deficiency		108 (50.9)	73 (90.1)	35 (26.7)	<0.001
Atrial fibrillation		83 (39.2)	30 (37)	53 (40.5)	0.620
Hypertension		163 (76.9)	64 (79)	99 (75.6)	0.564
Diabetes mellitus		66 (31.1)	31 (38.3)	35 (26.7)	0.078
Chronic kidney disease		25 (11.8)	16 (19.8)	9 (6.9)	0.005
Obstructive sleep apnea		25 (11.8)	9 (11.1)	16 (12.2)	0.809
Hyperlipidemia		81 (38.2)	29 (35.8)	52 (39.7)	0.571
Coronary artery disease		70 (33)	31 (38.3)	39 (29.8)	0.201
Peripheral artery disease		8 (3.8)	2 (2.5)	6 (4.6)	0.433
CVA/TIA		15 (7.1)	5 (6.2)	10 (7.6)	0.687
COPD		31 (14.6)	15 (18.5)	16 (12.2)	0.207
Depression		18 (8.5)	6 (7.4)	12 (9.2)	0.656
Malignancy		6 (2.8)	5 (6.2)	1 (0.8)	0.021
Echocardiographic data					
LVEF, %		57 ± 4.2	56.7 ± 4.1	57.3 ± 4.3	0.319
E/e’		11.3 ± 3.2	12.1 ± 3.6	10.9 ± 2.9	0.008
LA enlargement, %		126 (59.4)	55 (67.9)	71 (54.2)	0.048
LV concentric hypertrophy, %		131 (61.8)	53 (65.4)	78 (59.5)	0.391
Blood tests					
Hemoglobin, g/dL		12.8 ± 1.9	11 ± 1.1	13.9 ± 1.3	<0.001
Transferrin saturation, mean (%)		28.8	15.5	37	<0.001
Ferritin, ng/mL		94.5 (35–207)	35 (19.7–85)	141 (83.9–261)	<0.001
NT-proBNP, pg/mL		568.5 (278–1339)	1041 (373–1558)	430 (254–981)	0.002
BUN, mg/dL		18.8 ± 8.1	21.2 ± 9.4	17.3 ± 6.8	0.001
Serum Creatinine, mg/dL		0.9 ± 0.3	1 ± 0.5	0.9 ± 0.3	0.036
Serum Sodium, mmol/L		140.6 ± 10	141.2 ± 2.4	140.1 ± 12.6	0.437
Serum Potassium, mmol/L		4.5 ± 0.5	4.5 ± 0.5	4.5 ± 0.5	0.532
Serum Calcium, mg/dL		9.2 ± 0.5	9 ± 0.5	9.3 ± 0.5	<0.001
Uric acid, mg/dL		5.9 ± 1.4	6.2 ± 1.5	5.8 ± 1.4	0.053
Leukocyte, ×10^3^/µL		8 ± 2.3	8 ± 2.6	8 ± 2.2	0.913
C-reactive protein, mg/dL		4 (1.7–8)	3.6 (1.8–9.6)	4 (1.7–7)	0.055
TSH, µIU/mL		1.2 (0.8–1.9)	1.3 (0.7–1.9)	1.2 (0.8–1.9)	0.490
Medications, *n* (%)					
Angiotensin-converting enzyme inhibitor		56 (26.4)	23 (28.4)	33 (25.2)	0.607
Angiotensin receptor blocker		55 (25.9)	20 (24.7)	35 (26.7)	0.744
Beta-blocker		110 (51.9)	44 (54.3)	66 (50.4)	0.577
Aldosterone antagonists		29 (13.6)	9 (11.1)	20 (15.3)	0.392
Digoxin		18 (8.5)	6 (7.4)	12 (9.2)	0.656
Statin		55 (25.9)	22 (27.2)	33 (25.2)	0.751
Loop diuretic		45 (21.2)	20 (24.7)	25 (19.1)	0.332
Antiaggregant		68 (32.1)	32 (39.5)	36 (27.5)	0.068
Anticoagulant		71 (33.5)	27 (33.3)	44 (33.6)	0.970
Nonsteroidal anti-inflammatory drug		29 (13.7)	11 (13.6)	18 (13.7)	0.974
Oral antihyperglysemic		57 (26.9)	31 (38.3)	26 (19.8)	0.003
Insulin		13 (6.1)	6 (7.4)	7 (5.3)	0.543
Follow-up period, months		66.2 ± 12.1	57.8 ± 15.8	68.7 ± 11.3	<0.001
Long-term mortality, *n* (%)		60 (28.3)	44 (54.3)	16 (12.2)	<0.001

Data are presented as mean ± standard deviation or median and interquartile range or number (%). Abbreviations: BUN, blood urea nitrogen; COPD, chronic obstructive pulmonary disease; CVA, cerebrovascular accident; ECG, electrocardiogram; LA, left atrium; LV, left ventricle; LVEF, left ventricle ejection fraction; NT-proBNP, N-terminal pro-B-type natriuretic peptide; TSH, thyrotropin stimulating hormone; NYHA, New York Heart Association; TIA, transient ischemic attack.

**Table 4 diagnostics-14-00209-t004:** Clinical and laboratory features of patients with and without iron deficiency.

		Iron Deficiency (*n* = 108)	No Iron Deficiency (*n* = 104)	*p*-Value
Female sex, *n* (%)		64 (59.3)	52 (50)	0.176
Age, years		73.6 ± 9	67.6 ± 11.3	<0.001
NYHA	I–II	82 (75.9)	93 (89.4)	0.010
	III–IV	26 (24.1)	11 (10.6)	
Paroxysmal nocturnal dyspnea, *n* (%)		61 (56.5)	37 (35.6)	0.002
Bendopnea, *n* (%)		35 (32.4)	23 (22.1)	0.093
Reduced exercise tolerance, *n* (%)		103 (95.4)	98 (94.2)	0.708
Body mass index, kg/m^2^		28.6 ± 4.5	27.8 ± 4.7	0.234
Systolic blood pressure, mmHg		129.5 ± 15.4	129.7 ± 15.1	0.919
Diastolic blood pressure, mmHg		78.9 ± 9.5	78.6 ± 10.7	0.843
Heart rate, bpm		82 ± 16	82 ± 16	0.955
Jugular vein distention, *n* (%)		24 (22.2)	18 (17.3)	0.369
Peripheral edema, *n* (%)		25 (23.1)	15 (14.4)	0.105
Pulmonary crepitations, *n* (%)		10 (9.3)	3 (2.9)	0.053
ECG abnormality, *n* (%)		56 (51.9)	54 (51.9)	0.992
Comorbidities, *n* (%)				
Anemia		73 (67.6)	8 (7.7)	<0.001
Atrial fibrillation		46 (42.6)	37 (35.6)	0.295
Hypertension		82 (75.9)	81 (77.9)	0.735
Diabetes mellitus		36 (33.3)	30 (28.8)	0.481
Chronic kidney disease		15 (13.9)	10 (9.6)	0.335
Obstructive sleep apnea		10 (9.3)	15 (14.4)	0.244
Hyperlipidemia		40 (37)	41 (39.4)	0.721
Coronary artery disease		37 (34.3)	33 (31.7)	0.696
Peripheral artery disease		2 (1.9)	6 (5.8)	0.135
CVA/TIA		7 (6.5)	8 (7.7)	0.731
COPD		14 (13)	17 (16.3)	0.486
Depression		8 (7.4)	10 (9.6)	0.564
Malignancy		4 (3.7)	2 (1.9)	0.434
Echocardiographic data				
LVEF, %		57 ± 4.3	57 ± 4.2	0.766
E/e’		11.6 ± 3.5	11.1 ± 3	0.284
LA enlargement, %		70 (64.8)	56 (53.8)	0.104
LV concentric hypertrophy, %		69 (63.9)	62 (59.6)	0.522
Blood tests				
Hemoglobin, g/dL		11.9 ± 1.8	13.8 ± 1.4	<0.001
Transferrin saturation, mean (%)		18	40	<0.001
Ferritin, ng/mL		36.5 (19.9–65)	209.5 (133–314)	<0.001
NT-proBNP, pg/mL		726.5 (305–1346)	453 (261–1313)	0.098
BUN, mg/dL		19.4 ± 7.8	18.3 ± 8.5	0.321
Serum Creatinine, mg/dL		0.9 ± 0.3	0.9 ± 0.5	0.852
Serum Sodium, mmol/L		141.4 ± 2.3	139.8 ± 14.1	0.243
Serum Potassium, mmol/L		4.5 ± 0.5	4.5 ± 0.5	0.470
Serum Calcium, mg/dL		9.2 ± 0.5	9.3 ± 0.5	0.062
Uric acid, mg/dL		6.1 ± 1.4	5.8 ± 1.5	0.215
Leukocyte, ×10^3^/µL		7.9 ± 2.3	8.3 ± 2.5	0.234
C-reactive protein, mg/dL		3.5 (1.6–8)	4 (1.9–8)	0.960
TSH, µIU/mL		1.3 (0.8–1.9)	1.2 (0.8–1.8)	0.359
Medications, *n* (%)				
Angiotensin-converting enzyme inhibitor		30 (27.8)	26 (25)	0.647
Angiotensin receptor blocker		23 (21.3)	32 (30.8)	0.116
Beta-blocker		57 (52.8)	53 (51)	0.791
Aldosterone antagonists		12 (11.1)	17 (16.3)	0.267
Digoxin		8 (7.4)	10 (9.6)	0.564
Statin		26 (24.1)	29 (27.9)	0.527
Loop diuretic		23 (21.3)	22 (21.2)	0.980
Antiaggregant		36 (33.3)	32 (30.8)	0.689
Anticoagulant		41 (38)	30 (28.8)	0.160
Nonsteroidal anti-inflammatory drug		17 (15.7)	12 (11.5)	0.373
Oral antihyperglysemic		35 (32.4)	22 (21.2)	0.065
Insulin		6 (5.6)	7 (6.7)	0.721
Follow-up period, months		61.2 ± 15	69.4 ± 12	<0.001
Long-term mortality, *n* (%)		48 (44.4)	12 (11.5)	<0.001

Data are presented as mean ± standard deviation or median and interquartile range or number (%). Abbreviations: BUN, blood urea nitrogen; COPD, chronic obstructive pulmonary disease; CVA, cerebrovascular accident; ECG, electrocardiogram; LA, left atrium; LV, left ventricle; LVEF, left ventricle ejection fraction; NT-proBNP, N-terminal pro-B-type natriuretic peptide; TSH, thyrotropin stimulating hormone; NYHA, New York Heart Association; TIA, transient ischemic attack.

**Table 5 diagnostics-14-00209-t005:** Univariable and multivariable logistic regression analysis for long-term all-cause mortality.

	Univariable Analysis		Multivariable Analysis	
	HR (95% CI)	*p*-Value	HR (95% CI)	*p*-Value
Age, 1 year increase	1.103 (1.063–1.182)	<0.001	1.122 (1.039–1.261)	0.002
NYHA III–IV	3.101 (1.501–6.702)	0.002	1.401 (0.981–2.603)	0.058
PND	2.936 (1.603–5.004)	0.002	1.421 (1.097–2.508)	0.027
Cardiac murmur	2.401 (1.142–4.775)	0.010	2.115 (0.688–1.701)	0.241
Anemia	8.603 (6.401–9.990)	<0.001	5.401 (4.303–6.209)	0.001
Iron deficiency	6.073 (4.650–8.201)	<0.001	3.502 (2.204–6.701)	0.015
CKD	4.700 (1.971–6.002)	<0.001	2.166 (1.500–3.825)	0.020
Malignancy	0.180 (0.033–1.048)	0.037	0.225 (0.170–1.081)	0.326
LVEF, 1% increase	0.886 (0.824–0.987)	0.006	0.837 (0.725–0.974)	0.016
E/e’, 0.5 increase	1.105 (1.039–1.268)	0.007	1.064 (0.862–1.331)	0.550
NT-proBNP, 50 pg/mL increase	1.000 (1.000–1.001)	0.018	1.000 (0.999–1.000)	0.391
BUN, 0.1 mg/dL increase	1.045 (1.013–1.094)	0.011	0.914 (0.816–1.019)	0.098

Abbreviations: BUN, blood urea nitrogen; CI, confidence interval; CKD, chronic kidney disease; HR, hazard ratio; LVEF, left ventricle ejection fraction; NT-proBNP, N-terminal pro-B-type natriuretic peptide; NYHA, New York Heart Association; PND, paroxysmal nocturnal dyspnea.

## Data Availability

The data presented in this study are available on request from the corresponding author. The data are not publicly available due to privacy.

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
