# Peer review of "Anemia and Iron Deficiency Predict All-Cause Mortality in Patients with Heart Failure and Preserved Ejection Fraction: 6-Year Follow-Up Study"

_diagnostics, 2024, doi:10.3390/diagnostics14020209_

Round 1

Reviewer 1 Report

Comments and Suggestions for Authors

I reviewed with interest the manuscript by KöseoÄŸlu and Özlek "Anemia and iron deficiency predict all-cause mortality in patients with HFpEF: 6-year follow-up study".

The main issue that was examined in this study is the effect of anemia and iron deficiency on mortality in patients with CHF with preserved EF.

It should be recognized that although the study of the prognostic impact of anemia in patients with CHF is currently attracting sufficient attention from researchers, the group of patients with preserved LV ejection fraction has been less studied in this regard. In addition, this article examines not only the prognostic significance of anemia, but also iron deficiency, which allows us to obtain new scientific facts. However, during the review I had questions and comments to which I would like to receive answers from the authors.

1. The authors begin the introduction by citing the well-known classification of CHF (lines 40-47). This is unnecessary since this classification is widely known and used in routine clinical practice.

2. The article does not indicate the protocol number and the date of the local ethics committee.

3. In section 2.3. Statistical analyzes do not indicate on the basis of what criteria the indicators included in the multivariate logistic regression model were selected. The authors also did not indicate whether they took into account the internal correlation between the parameters included in this model. Failure to take such correlations into account may lead to incorrect results of multivariate analysis.

4. Although the authors tried to include recent publications on the prognostic impact of anemia in HFpEF, some sources still escaped their attention. For example, it would be advisable to consider the results of a meta-analysis on this topic in the article by Majmundar M, et al (Ref 1, see below). Also in the discussion it is possible to discuss the association of anemia with exercise tolerance in patients with HFpEF, which was studied in the article by Naito A, et al (Ref. 2, see below)

Refrences:

1.     Majmundar M, Doshi R, Zala H, Shah P, Adalja D, Shariff M, Kumar A. Prognostic role of anemia in heart failure with preserved ejection fraction: A systematic review and meta-analysis. Indian Heart J. 2021 Jul-Aug;73(4):521-523. doi: 10.1016/j.ihj.2021.06.011.

2.     Naito A, Obokata M, Kagami K, Harada T, Sorimachi H, Yuasa N, Saito Y, Kato T, Wada N, Adachi T, Ishii H. Contributions of anemia to exercise intolerance in heart failure with preserved ejection fraction-An exercise stress echocardiographic study. Int J Cardiol Heart Vasc. 2023 Aug 20;48:101255. doi: 10.1016/j.ijcha.2023.101255.

Comments on the Quality of English Language

No comments

Author Response

We would like to thank the editor and reviewers for their time and efforts in reviewing the manuscript. We also would like to thank the reviewers for their insightful comments on the paper. We have tried to make the recommended alterations suggested by the reviewers within our revised manuscript. Once again, we thank you for your kind interest and hope that the revisions and responses to the reviewer’s comments specified below satisfy the editorial board.

Response to Reviewer 1

Q1. I reviewed with interest the manuscript by KöseoÄŸlu and Özlek "Anemia and iron deficiency predict all-cause mortality in patients with HFpEF: 6-year follow-up study".

The main issue that was examined in this study is the effect of anemia and iron deficiency on mortality in patients with CHF with preserved EF.

It should be recognized that although the study of the prognostic impact of anemia in patients with CHF is currently attracting sufficient attention from researchers, the group of patients with preserved LV ejection fraction has been less studied in this regard. In addition, this article examines not only the prognostic significance of anemia, but also iron deficiency, which allows us to obtain new scientific facts.

A1. We appreciate the reviewer's thorough evaluation of our article and their positive comments on its scientific merit.

Q2. The authors begin the introduction by citing the well-known classification of CHF (lines 40-47). This is unnecessary since this classification is widely known and used in routine clinical practice.

A2. The relevant sentences have been deleted, and the following sentences have been added to the paragraph:

“The importance of HFpEF has become more apparent in recent years as diagnostic modalities for HFpEF have become increasingly clear, and awareness of this type of heart failure has increased among clinicians.”

Q3. The article does not indicate the protocol number and the date of the local ethics committee.

A3. The relevant sentence has been revised as follows:

“…all study protocols were approved by the institutional review board of Mugla Sitki Kocman University (approval date and number: 02.10.2023/94).”

Q4. In section 2.3. Statistical analyzes do not indicate on the basis of what criteria the indicators included in the multivariate logistic regression model were selected. The authors also did not indicate whether they took into account the internal correlation between the parameters included in this model. Failure to take such correlations into account may lead to incorrect results of multivariate analysis.

A4. The following sentences have been added in the relevant section:

“In the study, factors that showed a significant association with mortality between the deceased and survivor groups with a p-value less than 0.05 on univariate analysis were selected for inclusion in the multivariate analysis. This approach was taken to identify potential risk factors that may have contributed to the difference in mortality rates between the two groups. The goal was to determine which factors remained significant after accounting for potential confounding variables in the multivariate analysis.”

“The correlation between the parameters in the model was assessed by Pearson analysis.”

Q5. Although the authors tried to include recent publications on the prognostic impact of anemia in HFpEF, some sources still escaped their attention. For example, it would be advisable to consider the results of a meta-analysis on this topic in the article by Majmundar M, et al (Ref 1, see below). Also in the discussion it is possible to discuss the association of anemia with exercise tolerance in patients with HFpEF, which was studied in the article by Naito A, et al (Ref. 2, see below)

Refrences:

Majmundar M, Doshi R, Zala H, Shah P, Adalja D, Shariff M, Kumar A. Prognostic role of anemia in heart failure with preserved ejection fraction: A systematic review and meta-analysis. Indian Heart J. 2021 Jul-Aug;73(4):521-523. doi: 10.1016/j.ihj.2021.06.011.

Naito A, Obokata M, Kagami K, Harada T, Sorimachi H, Yuasa N, Saito Y, Kato T, Wada N, Adachi T, Ishii H. Contributions of anemia to exercise intolerance in heart failure with preserved ejection fraction-An exercise stress echocardiographic study. Int J Cardiol Heart Vasc. 2023 Aug 20;48:101255. doi: 10.1016/j.ijcha.2023.101255.

A5. We would like to express our gratitude to the reviewer for their valuable suggestions, which have strengthened the discussion of our research. We have added the following sentences to the discussion section and revised the references:

“In a comprehensive meta-analysis of 28,735 patients by Majmundar et al., anemia was found to be a prognostic factor for all-cause mortality and all-cause hospitalizations in patients with HFpEF [24].”

“The importance of long-term survival and its related factors in patient care cannot be overstated. However, it's worth noting that reduced exercise capacity can also significantly impact the quality of life for these individuals. Although our study did not directly measure this, the prevalence of heart failure symptoms and diastolic dysfunction in anemic patients suggests that this is a valid concern. A recent exercise stress echocardiography-based study by Naito et al. found that [28] anemic HFpEF patients experience more severe effects on exercise capacity and oxygen consumption compared to non-anemic patients, with respiratory inefficiency being particularly concerning during peak exercise.”

Reviewer 2 Report

Comments and Suggestions for Authors

Although the relationship between heart failure of all types, anemia and iron deficiency is well known, Koseoglu at all find a way to still make their article interesting. One minor suggestion to add the remark on the time interval from the moment of anemia diagnosis to the moment of death.

This study yields important solutions regarding incidence of iron deficiency in HFpEF pacients and all-cause mortality in this group. Furthermore, they find that anemic patients have a 5 times higher relative risk of death. It would be interesting if the authors offered an insight as to why the heart rate did not differ significantly between the 2 analysed groups.

Author Response

We would like to thank the editor and reviewers for their time and efforts in reviewing the manuscript. We also would like to thank the reviewers for their insightful comments on the paper. We have tried to make the recommended alterations suggested by the reviewers within our revised manuscript. Once again, we thank you for your kind interest and hope that the revisions and responses to the reviewer’s comments specified below satisfy the editorial board.

Response to Reviewer 2

Q1. Although the relationship between heart failure of all types, anemia and iron deficiency is well known, Koseoglu at all find a way to still make their article interesting. One minor suggestion to add the remark on the time interval from the moment of anemia diagnosis to the moment of death.

A1. We appreciate the reviewer's thorough evaluation of our article and their positive comments on its scientific merit. We also thank the reviewer for his/her valuable addition.

Our study, which employed a retrospective design, focused on patients who had been admitted to cardiology outpatient clinics between September 2017 and April 2018 and had received a diagnosis of HFpEF. We included patients whose admission and follow-up data were complete and analyzed their initial presentation data, as well as their mortality rates during follow-up. To determine the prevalence of anemia and iron deficiency, we relied on the initial diagnosis data. Unfortunately, since we did not have access to patient data prior to these dates, we were unable to pinpoint the exact onset of anemia or iron deficiency in the patients. Consequently, we cannot provide a clear timeframe from the first diagnosis of anemia or iron deficiency to death. However, our study found that patients with anemia and iron deficiency had significantly shorter follow-up periods than other patients due to their high mortality rates.

The following sentences have been added to the relevant sections:

“Patients without anemia had a longer follow-up period due to their lower mortality rate (68.7 vs. 57.8 months, p<0.001).”

“Patients with ID were observed for an average of 61.2 months, while those without ID were followed up for 69.4 months.”

Q2. This study yields important solutions regarding incidence of iron deficiency in HFpEF patients and all-cause mortality in this group. Furthermore, they find that anemic patients have a 5 times higher relative risk of death. It would be interesting if the authors offered an insight as to why the heart rate did not differ significantly between the 2 analysed groups.

A2. We thank the reviewer for his/her valuable contribution. We have added the following sentences to the discussion section and revised the references:

“It has been established that anemic patients experiencing chemoreceptor stimulation due to hypoxia tend to have higher heart rates as a result of sympathetic system activation [29]. However, our study revealed that there was no statistically significant difference in the average heart rate between anemic and non-anemic patients in our patient population. This outcome may be attributed to a number of factors, such as the relatively small sample size, the use of rate-decelerating treatment agents, and the absence of severe anemia (hemoglobin <7 g/dl) among the patients.”

Round 2

Reviewer 1 Report

Comments and Suggestions for Authors

I was not completely satisfied with the answer to my 3rd comment. The authors stated that “Correlation between model parameters was assessed using Pearson analysis.” However, the inclusion of variables that correlate with each other in a multiple logistic regression model leads to incorrect results of multivariate analysis. Authors should indicate which variables were correlated with each other and how this was taken into account in the logistic regression.

Comments on the Quality of English Language

No comments

Author Response

We would like to thank the editor and reviewers for their time and efforts in reviewing the manuscript. We also would like to thank the reviewers for their insightful comments on the paper. We have tried to make the recommended alterations suggested by the reviewers within our re-revised manuscript. Once again, we thank you for your kind interest and hope that the revisions and responses to the reviewer’s comments specified below satisfy the editorial board.

Response to Reviewer 1

Q1. I was not completely satisfied with the answer to my 3rd comment. The authors stated that “Correlation between model parameters was assessed using Pearson analysis.” However, the inclusion of variables that correlate with each other in a multiple logistic regression model leads to incorrect results of multivariate analysis. Authors should indicate which variables were correlated with each other and how this was taken into account in the logistic regression.

A1. We sincerely apologize for any confusion caused by our unclear response to the reviewer's question in the previous round. Upon further analysis, we conducted a multivariate logistic regression model by removing correlated parameters from the list to re-examine the significant p-values. Although the primary findings of our study remained consistent, we made necessary corrections to some of the results and added new findings to the tables. In addition, we have edited the numerical values in the main text accordingly. We appreciate the reviewer's valuable feedback, which helped us to address this important issue.

The following sentences have been added to the statistics section of the manuscript:

“Potential risk factors that were correlated with each other were not included in the multivariable logistic regression analysis. Since Hb is correlated with anemia, ferritin with ID, ankle swelling, and JVD with PND, these parameters were not included in the multivariate logistic regression analysis.